# Frequency, timing and risk factors for primary maternal cytomegalovirus infection during pregnancy in Quebec

Safari Joseph Balegamire[1,2], Christian Renaud[2,3], Benoît Mâsse[1,4], Kate Zinszer[1,5], Soren Gantt[2,3], Yves Giguere[6,7], Jean-Claude Forest[6,7], Isabelle Boucoiran[1,2,8]*

1 Department of Social and Preventive Medicine, École de Santé Publique de Université de Montréal, Montreal, QC, Canada, 2 Women and Children's Infectious Diseases Center, CHU Sainte-Justine Research Center, Montreal, Canada, 3 Department of Microbiology, CHU Sainte-Justine, Université de Montréal, Montréal, Canada, 4 Applied Clinical Research Unit, CHU Sainte Justine Research Center, Montreal, Canada, 5 Centre de recherche en santé publique, Université de Montréal, Montreal, Canada, 6 CHU de Québec-Université Laval Research Center, Quebec City, Canada, 7 Department of Molecular Biology, Medical Biochemistry and Pathology, Faculty of Medicine, Université Laval, Quebec City, Canada, 8 Department of Obstetrics and Gynecology, Division of Maternofetal Medicine, Université de Montréal, Montreal, Canada

* isabelle.boucoiran@umontreal.ca

**Data Availability Statement:** Data cannot be shared publicly because some data are confidential. Data are available from the CHU de Québec-Université Laval Ethics Review Board

## Abstract

### Introduction

Maternal Cytomegalovirus (CMV) infection in the first trimester (T1) of pregnancy is a public health concern, as it increases the risk of severe neurodevelopmental outcomes associated with congenital infection compared to infections occurring later during pregnancy.

### Objectives

To determine CMV seroprevalence in T1 of pregnancy, its trend, risk factors and the incidence rate of primary infection during pregnancy.

### Methods

Using the biobank of the prospective cohort "Grossesse en Santé de Québec" collected between April 2005 and March 2010 at the Québec-Laval Hospital, Québec, Canada, maternal CMV serology was determined using Abbott Architect Chemiluminescence microparticle immunoassays for immunoglobulin G(IgG), immunoglobulin M(IgM) titration and IgG avidity testing. Changepoint detection analysis was used to assess temporal trends. Risk factors associated with seropositivity were determined by multivariable logistic regression.

### Results

CMV seroprevalence in T1 of pregnancy was 23.4% (965/4111, 95% CI, 22.1–24.7%). The incidence rate for CMV primary infection during pregnancy was 1.8 (95% CI, 1.2–2.6) per 100 person-years. No changepoint was identified in the maternal CMV-seroprevalence trend. Multivariable analyses showed that T1 maternal CMV seropositivity was associated

(ethiquedelarecherche@chudequebec.ca) for researchers who meet the criteria for access to confidential data.

**Funding:** This work is funded by Canadian Institutes of Health Research (CIHR) Operating Grant (grand award number 386200). Dr Boucoiran is a recipient of a salary award (chercheur-boursier) from FRQ-S (Quebec's Health research fund). Dr Safari Joseph Balegamire is a recipient from a scholarship from Altona and FRQ-S.

**Competing interests:** The authors have no conflicts of interest to disclose.

with having one child OR 1.3 (95% CI, 1.10–1.73) or two or more children OR 1.5 (95%CI, 1.1–2.1), ethnicity other than Caucasian OR 2.1 (95% CI, 1.1–3.8) and country of birth other than Canada and the USA OR 2.8 (95% CI, 1.5–4.9).

## Conclusions

In this cohort, maternal seroprevalence in T1 of pregnancy and seroconversion rate were low. This information and identified risk factors could help guide the development and implementation of preventive actions and evidence-based health policies to prevent CMV infection during pregnancy.

## Introduction

Cytomegalovirus (CMV) is the most common congenital infection and a major cause of childhood disability worldwide. Although post-natal infection is largely benign, congenital CMV infection causes sequelae in approximately 20% of infected children, including deafness, blindness and neurodevelopmental delay [1]. Congenital CMV infection can occur either in primary maternal infection during pregnancy, or as a result of "non-primary" infection, which refers to either reactivation of a pre-existing latent CMV strain, or reinfection of the pregnant woman with a new viral strain [2, 3].

The adult seroprevalence of CMV varies widely between countries, from <50% to nearly 100% [4–6], and the prevalence of congenital CMV infection in a population is positively correlated with the CMV seroprevalence among women of childbearing age [7]. Thus, in low- and middle-income countries, where CMV infects >90% of the population during early childhood [7], non-primary maternal infection accounts for most congenital infections. In contrast, the prevalence of CMV infection is approximately 50% in most high-income countries, where a larger proportion of congenital cases are due to primary maternal infection during pregnancy.

The risk of congenital CMV infection associated with primary maternal infection during pregnancy has been estimated to be between 30 and 40% [7]. However, primary CMV infection during the first trimester of pregnancy is associated with more severe symptoms of congenital infection and worse neurological outcomes, compared to primary maternal CMV infection later in pregnancy [8–10]. Thus, determining maternal CMV seroprevalence and the rate of primary infection during pregnancy is essential for understanding the drivers of congenital CMV infection in a population. As few data are available from Canada, we undertook this study in Quebec to estimate the seroprevalence for CMV among pregnant women in the first trimester and the incidence of primary infection during pregnancy, as well as their associated risk factors.

## Materials and methods

### Study population

This is a secondary analysis using the biobank of the prospective cohort "Grossesse en Santé de Quebec", which took place from April 2005 to March 2010 at the Centre Hospitalier Universitaire (CHU) de Québec-Laval, Canada to study complications of pregnancy [11, 12]. A total of 7,855 women were enrolled at their first antenatal visit and provided samples and questionnaire data at enrollment and at delivery. The inclusion criteria for the cohort were age 18

years or older; gestational age of at least 10 weeks; and no chronic liver or kidney disease. Exclusion criteria were pregnancies with major fetal abnormalities and those ending in termination, miscarriage or fetal death before 24 weeks of gestation (n = 142). The population of interest for the current study consists of pregnant women who had available specimens at the first trimester (T1) and at delivery (T3), which represents 52% of the total Grossesse en Santé cohort (4,111/7,855)- see S1 Table. Relevant de-identified clinical and socio-demographic variables were extracted from the "Grossesse en Santé" database, including age, marital status, ethnicity, parity, level of education, annual household income, work status and country of birth.

Ethical approval for data collection and biobanking of the initial cohort, as well as use of the data and samples for subsequent studies, was granted by the Research Ethics Board of the Quebec-Laval Hospital. In addition, all aspects of this study were approved by the ethics committee of the CHU Sainte-Justine and the Science and Health Research Ethics Committee of the University of Montréal.

### Serological testing

Serological tests were performed on T1 and T3 maternal plasma samples at the CHU Sainte-Justine virology laboratory. All samples were tested using Abbott reagents of the same reference number. Plasma were tested for CMV IgG and, if positive, for CMV IgM, using the Abbott Architect Platform by Chemiluminescence microparticle immunoassays [13]. Health Canada-approved positivity limits are for IgG $\geq$6.0 AU/ml (arbitrary units per ml) and IgM $\geq$0.85 s/co (sample to cutoff). Baseline maternal CMV seropositivity was defined as the presence of a positive IgG result, regardless of IgG avidity results or IgM serologies. CMV IgG avidity testing was performed with the Abbott Architect when both IgG and IgM were positive on T1 samples to estimate the chronicity of infection (10). The thresholds for avidity were <50% (low avidity, indicating infected $\leq$4 months prior), between 50% and <60% (intermediate avidity, indeterminate timing), and $\geq$ 60% (high avidity, infected >4 months), as described by the manufacturer's instructions. Primary CMV infection was defined as seroconversion from IgG-negative to -positive from T1 to T3.

### Statistical analysis

Baseline CMV seroprevalence was defined as the proportion of participants who were CMV IgG seropositive in T1, out of the total number of participants. Changepoint detection analysis was performed to determine if there was a significant change in the trend of baseline seroprevalence over the study period [14]. A time series of monthly seroprevalence was created and the cpt.meanvar function in R software [14] was used for the changepoint detection.

Study data were subjected to bivariate analysis (= unadjusted analysis) using the Pearson chi-square test or Fisher exact test in order to assess associations between risk factors and maternal seropositivity and between risk factors and the incidence of primary CMV infection and then, to multivariable logistic regression (= adjusted analysis) analyses to examine adjusted associations between risk factors and maternal seropositivity.

The variables used in the multivariable logistic regression models to assess risk factors for CMV seropositivity in the first trimester were selected from the literature. They included age, marital status, ethnicity, parity, level of education, annual household income, work status and country of birth. The conditions applied to the logistic regression model were verified for goodness of fit using the Hosmer and Lemeshow test and the "linktest" to check whether the model specification was good. The absence of collinearity between the variables included in the model was verified by the variance inflation factors and the absence of extreme values (outliers) by graphical analysis of the scatter plot of residual deviance versus predicted values.

The incidence of primary CMV infection during pregnancy was determined by the number of primary infections between the T1 and T3 samples among participants that were seronegative at T1. This incidence rate was expressed out of 10000 person-days at risk. Uncontrolled analysis by confounding factors, i.e. unadjusted, was performed to determine the risk of primary infection by Incidence Rate Ratio. The small number of participants with primary infection precluded a multivariable analysis for this dependent variable.

Statistical analyses were performed with the statistical software STATA version 12.1 and R version 3.6.1

## Results

### Study cohort

The flow of the 4,111 study participants is illustrated in Fig 1. Most participants were Caucasian (96.7%), born in Canada or in the United States (96.2%).

The figure illustrates the flowchart and breakdown of the study population. During T1, 23% (n = 965) of women were CMV IgG positive. Of these, 113 were also IgM positive: 4 were low IgG avidity, 5 were intermediate avidity, and 104 were high avidity. Of the 3146 women participants who were IgG-negative at T1 (after excluding one patient with insufficient plasma from T3), 28 women were IgG positive at T3, indicating that they had a primary infection between T1 and T3.

### Seroprevalence (Fig 2)

The seroprevalence at T1 was 23.47% (965/4111; 95% CI 22.18–24.79%). Among seropositive patients who had both IgG and IgM positive in first trimester samples (n = 113), 4 seropositive

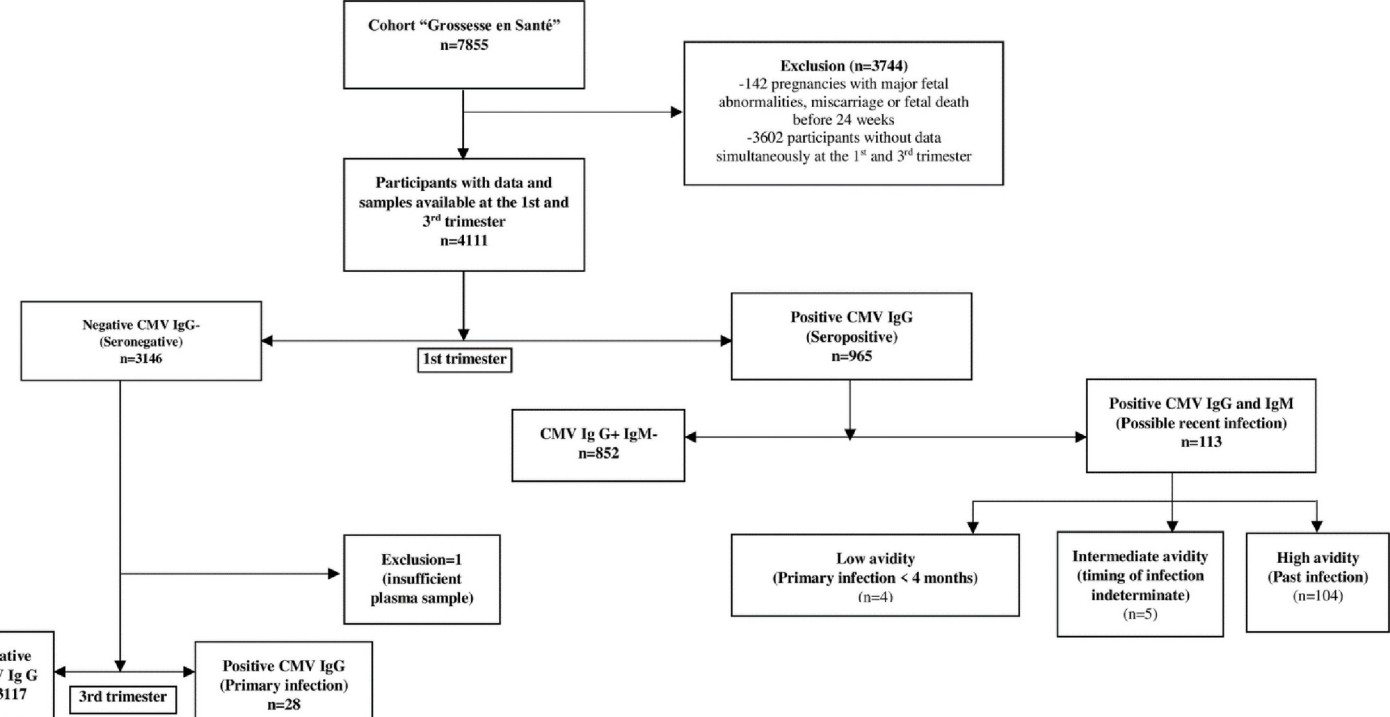

**Fig 1. Flowchart of study to detect cytomegalovirus seroprevalence and primary infection among pregnant women of the cohort "Grossesse en santé".**

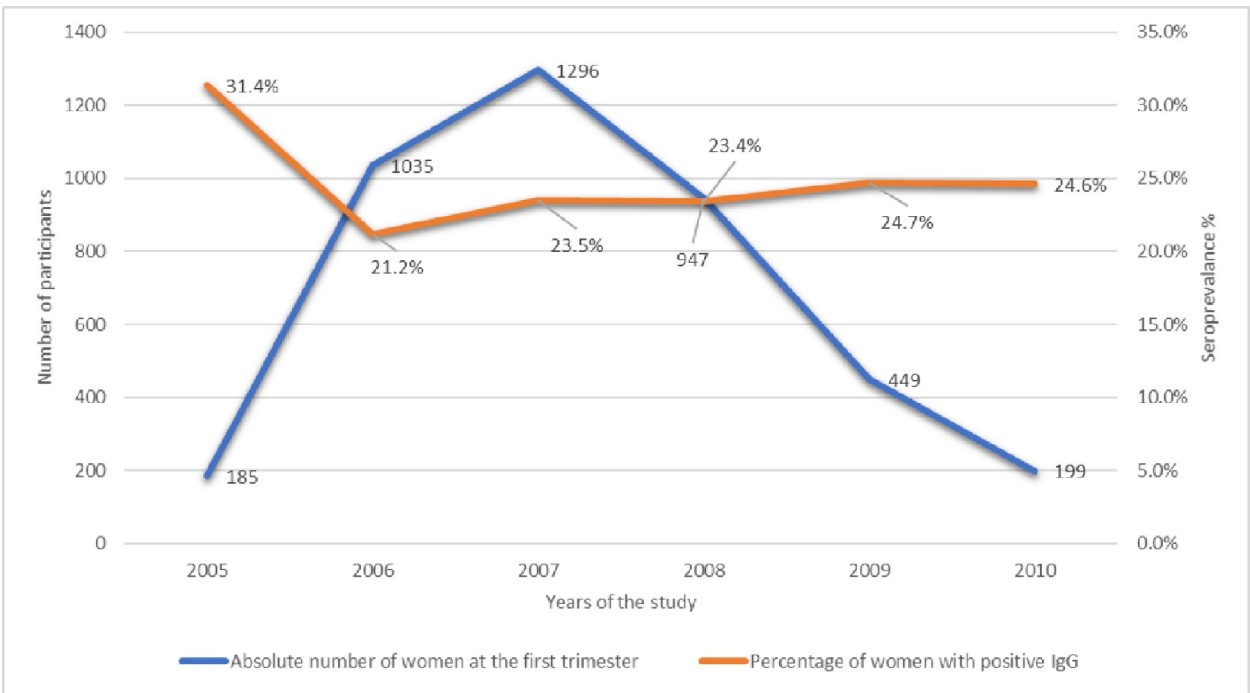

**Fig 2. Seroprevalence of CMV in the first trimester of pregnancy over year.**

patients had a low avidity, indicating infection ≤4 months prior, 5 had intermediate avidity and 104 had high avidity, indicating infection >4 months.

As shown in Fig 2, apart from the year 2005 when only 185 women had been enrolled in the cohort, annual IgG seroprevalence at T1 was stable and ranged between 21.2% and 24.7% over the duration of the study period.

No change in baseline maternal CMV seroprevalence over the period of study was detected with the changepoint detection analysis using the Binary Segmentation "Binseg" method with 'Exponential' as test statistic, and Schwarz Information Criterion "SIC" as penalty (Fig 3).

Fig 3 shows the trend of seroprevalence by CMV over the study period. No significant point of change in seroprevalence has been objectified during the study period presented by the straight line of the trend. Seroprevalence rates were analyzed on a monthly basis, with 0 being the study start month of April 2005 and 60 being the study end month of March 2010. The y-axis (0.10–0.35) represents the seroprevalence scale.

The risk factors for CMV seropositivity in the first trimester are shown in Table 1. In the multivariable analysis, only parity, ethnicity and country of birth remained significantly associated with baseline CMV seropositivity. Of note, among women born in Canada or in the USA, 98.1% were Caucasian, while among those born in other countries, 60.6% were Caucasian (P-value <0.001). This association did not affect the stability of the multivariable model.

## Primary CMV infection

As shown in Fig 1, among the 3145 pregnant women who were seronegative in T1 of pregnancy, 28 seroconverted, indicating a primary CMV infection rate of 0.9% (95% CI, 0.6–1.3%) during pregnancy. By adding the 4 cases who were seropositive- low avidity (recent infection in T1) to the 28 seroconverted cases, the incidence proportion of primary infection (32/3149) was 1.0% (95% CI, 0.7–1.4).

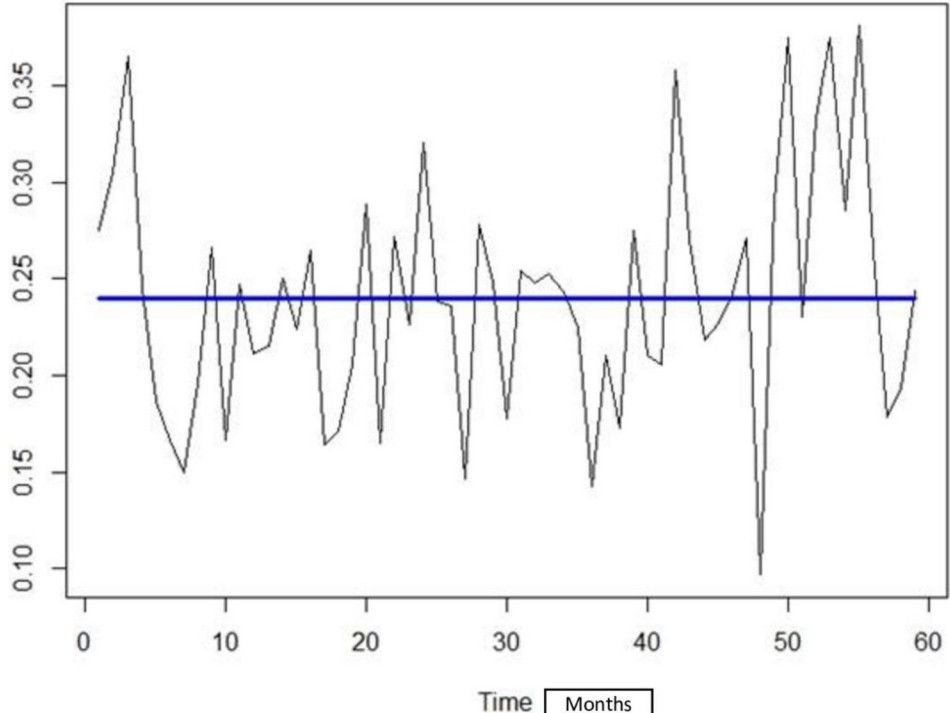

**Fig 3. Maternal CMV seroprevalence across all cohort participants from April 2005 to March 2010.**

The total incidence density of primary CMV infection during pregnancy was 1.8 (95% CI,1.9–2.7) per 100 person years or 0.5 (95% CI, 0.3–0.7) per 10,000 person days at risk. No variables were associated with the risk of primary infection in the unadjusted analysis except for parity. As demonstrated in Table 2, higher CMV incidence density was found in the following groups of women: age between 35 and 45 years, parity of at least one, annual household income ≥ $60,000, celibate, birth in Canada or the USA, having at least part-time work and having a college education.

## Discussions

In this prospective cohort study, we found a relatively low CMV seroprevalence in the first trimester (23.47%) and a low incidence rate of primary infection (1.86 per 100 person years). To our knowledge, this seroprevalence is the lowest reported to date. Previous Canadian studies reported estimated maternal CMV seroprevalences in Quebec of 40% [15] and 54% [16], and 55% in Alberta [17]. Previous reports from France, Europe, USA and Asia reported estimates that were all superior to 50% [4, 18]. The low estimate in our cohort could be explained by the almost homogeneous composition of the "Grossesse en Santé" cohort, with most participants being of Caucasian ethnicity and born in Canada or in the USA, whereas the composition of the previous Quebec cohort was more cosmopolitan [15, 19].

Parity, country of birth, and ethnicity were the risk factors significantly associated with maternal CMV seropositivity. Maternal seroprevalence of CMV in African-American and non-Hispanic black women is often reported to be higher than in Caucasian women [20–22]. In this study, most women who were born outside of Canada and the United States were from low- and middle-income countries, where the prevalence of CMV among children can reach 90% at age 2 years [23]. In contrast, in high-income countries, seroprevalence is lower and

**Table 1. Association between CMV IgG seropositivity and socio-demographic factors.**

| Variables | N total | n among IgG positive | % of IgG positive | Unadjusted | | Adjusted | | |
|---|---|---|---|---|---|---|---|---|
| | | | | OR* | IC 95% | OR | IC 95% | P-value |
| **Age** | 4111 | | | | | | | 0.6583 |
| 35–45 | 433 | 127 | 29.3 | 1.6 | 1.2—2.1 | 1.1 | 0.6—1.7 | |
| 30–34 | 1322 | 322 | 24.3 | 1.2 | 1.0—1.6 | 1.0 | 0.7—1.5 | |
| 25–29 | 1749 | 393 | 22.4 | 1.1 | 0.9—1.4 | 1.1 | 0.8–1.6 | |
| 18–24 | 607 | 123 | 20.2 | 1 | | 1 | | |
| **Parity** | 4111 | | | | | | | 0.006 |
| 2 children or more | 529 | 145 | 27.4 | 1.4 | 1.1–1.7 | 1.5 | 1.0–2.1 | |
| 1 child | 1610 | 402 | 24.9 | 1.2 | 1.0–1.4 | 1.3 | 1.1—1.7 | |
| 0 (nulliparous) | 1972 | 418 | 21.2 | 1 | | 1 | | |
| **Annual household income** | 3528 | | | | | | | 0.690 |
| Less than 15,499 $ | 124 | 38 | 30.6 | 1.5 | 1.0—2.2 | 0.9 | 0.4—1.6 | |
| 15,500 $- 24,999 $ | 201 | 55 | 27.3 | 1.2 | 0.9—1.7 | 0.9 | 0.5—1.6 | |
| 25,000 $- 39,999 $ | 466 | 116 | 24.8 | 1.1 | 0.8—1.4 | 1.1 | 0.8—1.5 | |
| 40,000 $- 59,999 $ | 777 | 166 | 21.3 | 0.9 | 0.7—1.1 | 0.8 | 0.6—1.1 | |
| 60,000 $ or above | 1960 | 445 | 22.7 | 1 | | 1 | | |
| **Ethnicity** | 3675 | | | | | | | 0.017 |
| Other* | 121 | 71 | 58.6 | 5.0 | 3.4—7.3 | 2.0 | 1.1—3.8 | |
| Caucasian | 3554 | 180 | 21.9 | 1 | | 1 | | |
| **Marital status** | 3793 | | | | | | | 0.1155 |
| Single | 263 | 62 | 23.5 | 1.0 | 0.7—1.4 | 1.4 | 0.9—2.2 | |
| Married | 837 | 225 | 26.8 | 1.2 | 1.0—1.5 | 0.8 | 0.6—1.0 | |
| Separated/Divorced | 26 | 6 | 23.5 | 1.0 | 0.4—2.6 | 1.1 | 0.2—4.2 | |
| Common-law partner | 2667 | 594 | 22.2 | 1 | | 1 | | |
| **Country of birth** | 3809 | | | | | | | <0.001 |
| Other countries | 144 | 95 | 65.9 | 6.9 | 4.8—9.9 | 2.8 | 1.5—4.9 | |
| Canada and USA | 3665 | 798 | 21.7 | 1 | | 1 | | |
| **Work** | 2573 | | | | | | | 0.795 |
| Yes | 2347 | 496 | 21.1 | 0.7 | 0.5—1.3 | 0.9 | 0.6—1.3 | |
| No | 226 | 60 | 26.5 | 1 | | 1 | | |
| **Level of education** | | | | | | | | 0.759 |
| None (high school not completed) | 147 | 52 | 35.3 | 1.8 | 1.2—2.6 | 1.3 | 0.7—2.5 | |
| Secondary (including professional) | 894 | 205 | 22.9 | 1.0 | 0.8—1.2 | 1.0 | 0.8—1,4 | |
| College (CEGEP) | 1280 | 296 | 23.1 | 1.0 | 0.8—1.2 | 1.0 | 0.8—1.3 | |
| University level | 1473 | 336 | 35.3 | 1 | | 1 | | |

*OR. Odds ratio

infections tend to occur later in life [23]. Women from non-Caucasian ethnicity born in Canada or in the USA may remain in a community with high CMV transmission rates, thereby increasing their risk of infection early in life [24, 25]. Given that parity is a proxy variable for the number of children at home, our finding is similar to previous reports [26] in that seroprevalence is positively correlated with the number of close child contacts (Table 1). Exposure to young children, who shed CMV at high viral loads in saliva and urine for prolonged periods when infected, is a known risk factor for CMV infection [27].

The primary infection rate in our study is similar to the estimates of annual seroconversion during pregnancy in other high-income countries, which range from 1 to 7% [28], but it is 3

**Table 2. Incidence of cytomegalovirus seroconversion in seronegative participants from the "Grossesse en santé" cohort study, Québec, 2005–2010 (n = 3145).**

| Variables | Person-time (days) | No of Seroconversion | Incidence Rate* | Unadjusted Incidence Rate Ratio (95%CI**) |
|---|---|---|---|---|
| **Age** | | | | |
| 35–45 | 53021 | 3 | 0.5 | 1.1 (0.1–7.0) |
| 30–34 | 173987 | 9 | 0.5 | 1.0 (0.3–4.8) |
| 25–29 | 236790 | 12 | 0.5 | 1.0 (0.3–4.5) |
| 18–24 | 84580 | 4 | 0.4 | 1 |
| **Parity** | | | | |
| 2 children or more | 66651 | 1 | 0.1 | 0.5 (0.01–3.8) |
| 1 child | 209993 | 19 | 0.9 | 3.0 (1.2–8.1) |
| 0 (nulliparous) | 271734 | 8 | 0.2 | 1 |
| **Annual household income** | | | | |
| Less than 15,499 $ | 14,600 | 0 | 0 | - |
| 15 500 $ - 24 999 $ | 25 239 | 1 | 0.3 | 0.6 (0.01–3.9) |
| 25 000 $ - 39 999 $ | 60 535 | 1 | 0.1 | 0.2 (0.01–1.6) |
| 40 000 $ - 59 999 $ | 105 841 | 3 | 0.2 | 0.4 (0.1–1.5) |
| 60 000 $ or above | 265110 | 17 | 0.4 | 1 |
| **Ethnicity** | | | | |
| Other (African Canadian, Asian, Latin-Canadian, Canadian First Nation and other) | 8557 | 1 | 1.1 | 2.5 (0.1–15.8) |
| Caucasian | 482 806 | 22 | 0.4 | 1 |
| **Marital status** | | | | |
| Single | 34 652 | 3 | 0.8 | 1.9 (0.3–6.8) |
| Married | 105946 | 6 | 0.5 | 1.2 (0.4–3.4) |
| Separated/Divorced | 3361 | 0 | 0 | - |
| Common-law partner | 361926 | 16 | 0.4 | 1 |
| **Country of birth** | | | | |
| Other countries | 8569 | 0 | - | - |
| Canada or USA | 499104 | 21 | 0.5 | 1 |
| **Work (part-time or full-time)** | | | | |
| Yes | 322672 | 16 | 0.4 | - |
| No | 28841 | 0 | - | - |
| **Level of education** | | | | |
| None (high school not completed)3 | 16426 | 0 | - | - |
| Secondary (including professional)1 | 119393 | 7 | 0.5 | 1.6 (0.4–5.5) |
| College (CEGEP) | 170864 | 11 | 0.6 | 1.8 (0.6–5.5) |
| University level | 199061 | 7 | 0.3 | 1 |

* /10.000 person-days at risk;

** CI = confidence interval

times lower than the seroconversion estimated at 1.4 per 10,000 person-days in the previous Quebec study from Lamarre et al. [15]. Of note, the proportion of women with low avidity rate (3.5%) among Ig M-positive women in our cohort was low compared to other studies where the rates of low avidity were found to be greater than 20% [29–31]. These findings support the positive association between baseline CMV seroprevalence in a population and the rate of virus transmission in this same population.

Compared to studies conducted in Canada and other settings, this large prospective study allowed for precise determination of first trimester maternal CMV seroprevalence and the

incidence of primary infection in pregnant women. However, this study has some limitations. First, the "Grossesse en Santé" cohort may not be generalizable to the entire Quebec population, as they tended to come from a high socioeconomic level (see Table 1) and therefore may underestimate maternal CMV seroprevalence in the wider Québec population [22, 32, 33]. This could also explain the lack of significant association between low socioeconomic status and CMV seropositivity in our study. Commonly-used indicators for measuring social status are education, occupation, and income, although these are limited in different contexts [34]. In our database, only education and income were available, and these two factors are often associated proportionately [35]. Second, the number of non-Caucasian subjects was small compared to Caucasians and may not be representative. Third, the small number of participants with primary infection in the cohort precluded a multivariate analysis to determine the risk factors associated with primary infection. This also precluded time series analyses of seroconversion rates over time. Fourth, some variables of interest, such as risk factors for CMV, were not present in the "Grossesse en Santé" cohort database, including occupation or preventive job withdrawal or reassignment job which is recommended in Quebec for prevention of CMV infection [36]. Finally, the rate of congenital CMV infection in this cohort was not available.

## Conclusions

In conclusion, the low first trimester CMV seroprevalence and low frequency of CMV primary infection found in this study, when compared to estimates from a different cohort from Quebec [15, 19], show that first trimester CMV seroprevalence and primary infection incidence rate are heterogeneous within the province of Quebec. The identified risk factors should be considered when devising interventions to prevent congenital infection. In addition, despite the low maternal CMV seroprevalence, the results of this study show that maternal CMV infection during pregnancy remains a public health priority in order to prevent congenital CMV infection. Hygiene precautions and eventually vaccination have the potential to prevent maternal and congenital CMV infection but may also have differential efficacy against primary and non-primary infections. Thus, in the future, it will be important to conduct studies to regularly assess the maternal incidence and seroprevalence of CMV infection on a longitudinal basis using large population-based samples, and to assess the risk factors for infection among different ethnic groups.

## Supporting information

**S1 Table. Distribution of clinical and sociodemographic characteristics in the study participants compare to the whole "grossesse en santé" cohort.**
(DOCX)

**S1 File.**
(DOCX)

**S2 File.**
(XLSX)

**S3 File.**
(XLSX)

## Acknowledgments

The authors thank Dr. Christian Band for his critical appraisal and editing of the manuscript.

## Author Contributions

**Conceptualization:** Safari Joseph Balegamire, Christian Renaud, Benoît Mâsse, Isabelle Boucoiran.

**Data curation:** Yves Giguere, Jean-Claude Forest.

**Formal analysis:** Safari Joseph Balegamire, Christian Renaud, Benoît Mâsse, Kate Zinszer.

**Methodology:** Safari Joseph Balegamire, Christian Renaud, Benoît Mâsse, Isabelle Boucoiran.

**Project administration:** Isabelle Boucoiran.

**Supervision:** Benoît Mâsse, Isabelle Boucoiran.

**Writing – original draft:** Safari Joseph Balegamire, Benoît Mâsse, Isabelle Boucoiran.

**Writing – review & editing:** Safari Joseph Balegamire, Christian Renaud, Benoît Mâsse, Kate Zinszer, Soren Gantt, Yves Giguere, Jean-Claude Forest, Isabelle Boucoiran.

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
