## [Decision Letter · Decision Letter 0]

12 Feb 2021

PONE-D-20-40657

Frequency, timing and risk factors for primary maternal cytomegalovirus infection during pregnancy in Quebec.

PLOS ONE

Dear Dr. Boucoiran,

Thank you for submitting your manuscript to PLOS ONE. After careful consideration, we feel that it has merit but does not fully meet PLOS ONE’s publication criteria as it currently stands. Therefore, we invite you to submit a revised version of the manuscript that addresses the points raised during the review process.

Your manuscript has been evaluated by two expert reviewers whose comments are appended below.  In general, they felt the paper was well written, but attention to several details is necessary in order for the manuscript to be publishable.  In particular, the figures require legends and axis labels and tables require more clarification, as noted by both reviewers.  In addition, reviewer #2 asks for the inclusion of primary data from the IgG and IgM tests, and I agree that these data are necessary to better analyze and contextualize the results.  Please address all of the reviewer comments below in your rebuttal letter when you submit the revised manuscript.

We look forward to receiving your revised manuscript.

Kind regards,

Juliet V Spencer, Ph.D.

Academic Editor

PLOS ONE

Journal Requirements:

2.We note that you have indicated that data from this study are available upon request. PLOS only allows data to be available upon request if there are legal or ethical restrictions on sharing data publicly. For information on unacceptable data access restrictions, please see http://journals.plos.org/plosone/s/data-availability#loc-unacceptable-data-access-restrictions.

Reviewers' comments:

Reviewer's Responses to Questions

**Comments to the Author**

1. Is the manuscript technically sound, and do the data support the conclusions?

Reviewer #1: Yes

Reviewer #2: Yes

2. Has the statistical analysis been performed appropriately and rigorously? 

Reviewer #1: Yes

Reviewer #2: Yes

3. Have the authors made all data underlying the findings in their manuscript fully available?

Reviewer #1: No

Reviewer #2: Yes

4. Is the manuscript presented in an intelligible fashion and written in standard English?

Reviewer #1: Yes

Reviewer #2: Yes

5. Review Comments to the Author

Reviewer #1: In the here presented manuscript by Balegamire et al. the authors examine the seroprevalence of expecting mothers in Montréal and compare their serostatus in the first trimester to their serostatus at delivery to determine the prevalence of primary maternal CMV infection during pregnancy. This data can be used to infer the number of congenital CMV infections and by that the number of newborns with potentially lifelong sequelae in the studied target population. The presented study is well written and the authors present their data overall in a very nice and understandable manner. The conclusions of the manuscript are fully in-line with the presented results and the authors do discuss the limitations of their approach and studied group of individuals. Furthermore, this data clearly points out that there is a sizeable vulnerable population of women of childbearing age in Montréal and the detection of primary infections in CMV negative expecting mothers is a clear indication that some of these infection will results in affected children that will required follow up care, potentially for the rest of their lives. Nevertheless, it has to be stated that the presented results are not entirely novel as similar data resulting in similar conclusion have been generated previously by multiple groups around the globe. Yet, the presented study is scientifically sound and might be of interest to researchers and physicians in the field and should be deemed worth publishing. There are a few minor corrections the authors should make to the manuscript before publication to help the reader better understand the presented results.

1) For Table 1 it might be helpful if the authors could explain in the text or the materials and methods section what their abbreviations on top of each row mean and what the difference between their unadjusted and adjusted analysis is.

2) Similarly, for table 2 further explanations on what the rate and what the unadjusted rate is are needed. Also, there are * and ** after Rate and Unadjusted RR, but I can’t find what these asterisks refer to.

3) I can’t find figure legends. Figure 1 is probably fine, but figure 2 and 3 might benefit from some figure legends explaining what is shown.

4) There are some formatting issue with missing gaps or additional full stops in lines 135, 201, 214, 222, 224 and 232

Reviewer #2: To the authors of PONE-D-20-40657

SF Balegamire et al submitted a manuscript entiteled

Frequency, timing and risk factors for primary maternal cytomegalovirus infection

during pregnancy in Quebec.

To PLOS One.

The authors report on CMV seroprevalence in the first trimester (T1) in pregnancy and risk factors for the rate of CMV primary infection in a cohort of Caucasian ethnicity in the Quebec Region.

In a retrospective study using cryopreserved serum samples of the “Grossesse en Sante de Quebec” in total N=7855 pregnant women were included. About half of this cohort were excluded (?, N=3744). From the remaining 4111 participants data from T1 and T3 (third trimester) were available, to observe potential seroconversion of seronegative pregnant women from T1 to T3. Seroconversion was defined by conversion from CMV IgG negative to CMV IgG positive. In T1, 3146 individuals were CMV seronegative and in T3, 28 cases of maternal seroconversions could be documented. This correlates with 0,9% of maternal seroconversion. In contrast, in the T1 seropositive cohort of 965 pregnant women, 113 showed CMV IgG+ IgM+, which were attributed to “possible recent infection”.

In total, 104 pregnant women were infected latently and 4 primary infections of the CMV IgG+ IgM+ cohort were defined by low CMV IgG avidity. Multivariate analysis revealed parity, ethnicity and country of birth as risk factors.

The results are discussed in detail and the paper contributes seroepidemiological data of an additional cohort of high income from a developed country.

However, there are some open questions with regard to virological methods.

1.) The authors should please add original data from their seroepidemiological study, like IgM indices, IgG levels and IgG avidity assays. This could help to understand the different subcohorts of the seropositive pool of 965 mothers.

2.) Why in the seroconversion data in T3 of primary infections no CMV IgG avidity data are given? It would also be helpful to see the original data of CMV IgG levels, IgM indices, and CMV IgG avidity.

3.) These lacking data should-if available- also be evaluated statistically to characterise CMV primary infections versus non-primary infections or latent CMV infections. Do all primary infected women in T3 really express clear low IgG avidity?

4.) Fig 2: how the drop in CMV seroconversion from 2005 to 2006 can be explained? Is it a potential bias, since in 2005 only 185 pregnant women were included?

6. PLOS authors have the option to publish the peer review history of their article (what does this mean?). If published, this will include your full peer review and any attached files.

Reviewer #1: No

Reviewer #2: **Yes: **Klaus Hamprecht

---

## [Author Response · Author response to Decision Letter 0]

12 May 2021

REVIEWER COMMENTS AND ACTIONS TAKEN

Reviewer #1: 

1) For Table 1 it might be helpful if the authors could explain in the text or the materials and methods section what their abbreviations on top of each row mean and what the difference between their unadjusted and adjusted analysis is.

The column headings in Table 1 have been clarified. Details regarding unadjusted and adjusted analyses have been provided in “Materials and Methods” (see lines 136-141, 151-154,).

2) Similarly, for table 2 further explanations on what the rate and what the unadjusted rate is are needed. Also, there are * and ** after Rate and Unadjusted RR, but I can’t find what these asterisks refer to.

The column headings in table 2 have been modified for clarity and asterisks are defined at the bottom of the table. The unadjusted rate ratio is explained in “Materials and Methods” (lines 152 – 154).

3) I can’t find figure legends. Figure 1 is probably fine, but figure 2 and 3 might benefit from some figure legends explaining what is shown.

Legends to the Figures have been added (see Results).

4) There are some formatting issues with missing gaps or additional full stops in lines 135, 201, 214, 222, 224 and 232

Formatting issues have been addressed.

Reviewer #2 : 

1) The authors should please add original data from their seroepidemiological study, like IgM indices, IgG levels and IgG avidity assays. This could help to understand the different subcohorts of the seropositive pool of 965 mothers.

Original data from our seroepidemiological study data (IgM indices, IgG levels and IgG avidity assays) have been included as a part of this resubmission (see attached excel files of data: 1) extraction projet PR67F Ig G et Ig M and 2) PR67F 2019-04-26 avidite).

2) Why in the seroconversion data in T3 of primary infections no CMV IgG avidity data are given? It would also be helpful to see the original data of CMV IgG levels, IgM indices, and CMV IgG avidity.

The purpose of Ig G avidity analysis is to distinguish between recent and past infection. T3 blood sample analyses enabled us to establish the frequency of primary infection occurring through IgG seroconversion between T1 and T3. The authors feel that additional avidity analyses are not warranted. 

3) These lacking data should-if available- also be evaluated statistically to characterise CMV primary infections versus non-primary infections or latent CMV infections. Do all primary infected women in T3 really express clear low IgG avidity?

We did not have data for T3 CMV IgG avidity to characterize primary versus other CMV infection. Rather, we used IgG seroconversion, which is a better indicator of recent infection than avidity. Standard serological testing and avidity cannot distinguish for non-primary infection.

4) Fig 2: how the drop in CMV seroconversion from 2005 to 2006 can be explained? Is it a potential bias, since in 2005 only 185 pregnant women were included?

We believe Reviewer 1 is referring to seroprevalence rather than seroconversion. We do not know why seroprevalence was higher in 2005 than during the rest of the study duration and have changed our manuscript to reflect this (lines 180-182). Irrespective of this “outlier” changepoint analysis revealed no observable trends in seroprevalence over the study, including between years 2005 and 2006.

---

## [Editor Report · Decision Letter 1]

14 May 2021

Frequency, timing and risk factors for primary maternal cytomegalovirus infection during pregnancy in Quebec.

PONE-D-20-40657R1

Dear Dr. Boucoiran,

We’re pleased to inform you that your manuscript has been judged scientifically suitable for publication and will be formally accepted for publication once it meets all outstanding technical requirements.

Kind regards,

Juliet V Spencer, Ph.D.

Academic Editor

PLOS ONE
---

## [Editor Report · Acceptance letter]

16 Jun 2021

PONE-D-20-40657R1 

Frequency, timing and risk factors for primary maternal cytomegalovirus infection during pregnancy in Quebec. 

Dear Dr. Boucoiran:

I'm pleased to inform you that your manuscript has been deemed suitable for publication in PLOS ONE. Congratulations! Your manuscript is now with our production department. 

Kind regards, 

on behalf of

Dr. Juliet V Spencer 

Academic Editor

PLOS ONE